# Inducing programmed cell death trough MazEF system to combat *Staphylococcus aureus*: A non-antibiotic treatment candidate

**Shahriar Bakhti, Mohammad Hossein Ahmadi** ORCID **\*, Parviz Owlia**

Department of Microbiology, Faculty of Medicine, Shahed University, Tehran, Iran

\* mhahmadi@shahed.ac.ir

**Data Availability Statement:** All relevant data are within the manuscript and its Supporting Information files.

## Abstract

The aim of the present study was to evaluate the role of extracellular death factor (EDF) derived from *Escherichia coli* in the induction of programmed cell death (PCD) in methicillin-resistant and -susceptible *Staphylococcus aureus* (MRSA and MSSA). The confirmation of bacterial strains as well as the minimum inhibitory concentration (MIC) test were performed according to CLSI, 2022. The extraction and efficacy determination of EDF as well as the CFU assessment were done. The expression of *mazE* and *mazF* gens in different conditions was evaluated by Real-time PCR. The likely formation of persister cells from MRSA and MSSA, and the possible synthesis of EDF in old cultures of these pathogens was evaluated, as well. The combination of EDF of two *E. coli* strains and sub-MIC rifampin reduced the CFUs of MRSA and MSSA strains in mid-logarithmic growth phase while increased the expression of *mazF* several times more than *mazE* gene. The expression of these genes in different conditions were unlike. EDF was produced in the old cultures of MRSA and MSSA. The supernatant of *E. coli* 25922 was more powerful than the clinical strain ones to decrease the CFUs of the MRSA and MSSA. The EDF derived from *E. coli* in combination with sub-MIC rifampin could induce PCD in MRSA and MSSA through activation of the MazEF system. This phenomenon could be exploited as a non-antibiotic treatment candidate to combat the infections caused by the antibiotic-resistant pathogens. However, more studies should be performed in this regard.

## Introduction

*Staphylococcus aureus* is one of the key pathogens that can cause many healthcare problems especially in immunocompromised people. Although this bacterium may present as a component of skin and mucous membranes as normal microbiota, Methicillin-Resistant *Staphylococcus aureus* (MRSA) and Vancomycin-Resistant *Staphylococcus aureus* (VRSA) strains could be very dangerous and create high range of morbidity and mortality. Unfortunately, there is an enhanced resistance of *S. aureus* to many different antibiotics causing many infections difficult to treat, including bacteremia, carbuncles, cellulitis, scalded skin syndrome, osteomyelitis,

**Funding:** This work is based upon research funded by Iran National Science Foundation (INSF) under project No. 4002145. The funders had no role in study design, data collection and analysis, decision to publish, or preparation of the manuscript.

**Competing interests:** The authors have declared that no competing interests exist.

infective endocarditis, septic arthritis, prosthetic device infections, pulmonary infections, pneumonia, meningitis, toxic shock syndrome, and urinary tract infections [1, 2].

Toxin-Antitoxin (TA) system has been identified about four decades ago. TA systems consist of two components; a stable toxin and an unstable antitoxin. The toxins are almost protein and antitoxins are either RNA or protein. In normal conditions, the toxins are always neutralized and inhibited by the antitoxins but in stressful conditions, the unstable antitoxins will be broken and consequently the stable toxins could destroy the cells. Today, TA systems are typically classified to more than eight groups according to how the antitoxin neutralizes the toxin. Among them, one of the important systems is type two (type II) toxin-antitoxin system that the antitoxin protein binds and neutralizes the toxin protein, post-translationally. Within this class, the best studied system is MazEF toxin antitoxin system in which MazE is a labile antitoxin and MazF is a stable toxin encoded by *mazE* and *mazF* genes, respectively. MazE prevents the lethal effect of MazF [3–5].

Importantly, MazEF toxin-antitoxin is regulating system responsible for bacterial programmed cell death (PCD). The stressful conditions such as inhibition of transcription and/or translation by antibiotics (for example rifampin), inhibition of *mazEF* transcription by overproduction of ppGpp and DNA damage, could prevent the expression of the *mazEF* operon leading to the activation of the toxin (Maz F) and subsequently the PCD would be occurred [6].

PCD induced by MazEF toxin-antitoxin system (*mazEF*-mediated cell death) is occurred among bacterial communities as a population based phenomenon that needs a differential quorum-sensing (QS) peptide called the extracellular death factor (EDF). The EDF of *Escherichia coli* is a liner peptide consisting of five amino acids including Asn-Asn-Trp-Asn-Asn-OH (NNWNN) that is produced by the enzyme glucose-6-phosphate dehydrogenase. It has been shown that the formation of MazEF complex is inhibited by EDF in stressful conditions. This inhibition increases the endoribonucleolytic activity of MazF causing the mazEF-mediated cell death. Surprisingly, some studies has shown that an EDF extracted from a bacterial species (also a synthetic EDF) may induce MazEF-mediated cell death in the other bacterial species (the interspecies bacterial cell death) [6–9].

The aim of the present study was to extract the *E. coli* EDF and evaluate its effect in the induction of programmed cell death due to MazEF toxin-antitoxin system in MRSA and MSSA strains in different conditions. Moreover, the possible production of EDF in old cultures of MRSA and MSSA as well as the likely formation of persister cells from these pathogens were also investigated.

## Materials and methods

### Study design

Ethics approval for this study was obtained from the Research Ethics Committee of Shahed University (Approval ID: IR.SHAHED.REC.1400.037). This prospective study was conducted between January 2022 and December 2023. The patient from which a clinical strain of *E. coli* was isolated provided written informed consent.

### Identification and validation of bacterial strains

In the present study, three standard strains as well as one clinical strain have been used as following: *S. aureus* ATCC43300 as a standard strain of MRSA, *S. aureus* ATCC25923 as a standard strain of MSSA, *E. coli* ATCC25922, and a clinical strain of *E. coli* isolated from a patient with urinary tract infection.

Conventional culture media (such as Blood agar, MacConkey agar, TSI, Luria-Bertani (LB) agar, and LB broth) were used for identification of the bacterial strains. Identification and confirmation methods were applied to specify and verify the *S. aureus* (MRSA and MSSA) and *E. coli* strains according to CLSI 2022 standard guideline [10]. The liquid M9 minimal medium with 1% glucose and a mixture of amino acids (10 μg/ml of each) were used to culture the *E. coli* strains and extract the EDF [6].

## Determination of minimal inhibitory concentration (MIC)

The MIC of rifampin for MRSA and MSSA strains was determined by macro broth dilution method [10].

## EDF extraction

The supernatants of the *E. coli* strains that served as an EDF donor (ATCC25922 and the clinical isolate) were separated as EDF using the following method: A culture of the *E. coli* strain was grown in M9 medium with 1% glucose and a mixture of amino acids (10 μg/ml of each) with shaking (160 rpm) at 37°C for 12 h (as the standard EDF). The cells were diluted 1:100 in M9 medium and were grown with shaking (160 rpm) at 37°C to mid-logarithmic growth phase (optical density at 600 nm [OD600] of 0.6; $2.5*10^8$ cells/ml). Cells were then centrifuged at 14,000 rpm for 5 min. The supernatant was collected and filtered through a 0.22-μm filter and the filtrates were stored at 4°C [6]. Furthermore, the *E. coli* strains were cultured in M9 with 1% glucose and a mixture of amino acids (but not to mid-logarithmic growth phase), in LB agar, M9 without glucose, and M9 without amino acids to compare the efficacy of their supernatants with our standard EDF.

## Determination of the EDF efficacy

Initially, MRSA or MSSA was grown in LB broth overnight. Then they were diluted, first by pre-warmed LB broth to a density of $3*10^4$ cells/ml and incubation at 37°C without shaking for 40 minutes (as the dense culture), second by pre-warmed LB broth to a density of $3*10^4$ cells/ml and incubation at 37°C without shaking for 20 minutes and another 20 minutes with added 2 μg/ml rifampin at 37°C without shaking (to assess only rifampin efficacy), third by supernatants of each *E. coli* strains to a density of $3*10^4$ cells/ml for 20 minutes and another 20 minutes with added 2 μg/ml rifampin at 37°C without shaking (to assess the efficacy of *E. coli* EDF plus rifampin), and fourth by supernatants of each *E. coli* strains to a density of $3*10^4$ cells/ml for 40 minutes (to assess the efficacy of only *E. coli* EDF). Then loss of viability was determined by counting the Colony Forming Units (CFUs) [8].

Moreover, in addition to mid-logarithmic growth phase (OD = 0.6), the supernatant of *E. coli* 25922 was also collected in the following conditions: in OD = 0.28, OD = 0.4, OD = 0.78, OD = 1.3, in LB broth (24h and 72h), M9 without glucose, and M9 without amino acids. Then the effect of these different supernatants on CFUs of MRSA and MSSA was assessed accordingly.

## Evaluation of *mazE* and *mazF* genes expression

The expression of *mazE* and *mazF* genes in MRSA and MSSA was evaluated by a Real-time PCR method. For this, the MRSA and MSSA strains were cultured in LB broth overnight. Then, they were diluted as mentioned for the pervious method (Determination of the EDF efficacy). For this purpose, MRSA and MSSA were affected by sub-MIC rifampin (2 μg/ml) alone, EDF (of mid-logarithmic growth phase, OD = 0.6) alone, and EDF plus rifampin. The RNAs

**Table 1. The specific primers designed and used to target *mazE*, *mazF*, and 16s rRNA genes.**

| Length (bp) | Sequence (5'→3') | Strand | Tm (°C) |
|---|---|---|---|
| 16s rRNA Primers | | | |
| 20 | GCTTGCTTCTCTGATGTTAG | Forward | 58 |
| 20 | AGACCGTCTTTCACTTTTGA | Reverse | 58 |
| *mazE* Primers | | | |
| 20 | GTCAAAATAGAAGTCATAGC | Forward | 58 |
| 18 | GTTCGCTAGGGAGAGATT | Reverse | 55 |
| *mazF* Primers | | | |
| 18 | GCGAAAATACCGACACAT | Forward | 56 |
| 21 | GCATTCAGCCCTAAACTAATC | Reverse | 58 |

of all samples (from each condition) were extracted using Hybrid-R™ RNA Extraction Kit (GeneAll, South Korea). Afterward, cDNA of all samples were synthesized using Easy™ cDNA Synthesis Kit (Pars Tous, Iran). The specific forward and reverse primers for *mazE*, *mazF* and 16s rRNA genes (Table 1) were designed using a primer design software (Vector NTI version 5.0 Informax, Inc., North Bethesda, MD, USA) and the primers specificity was checked using the Primer-BLAST tool of National Center for Biotechnology Information (NCBI). Finally, the expression of the *mazE* and *mazF* genes in these conditions was evaluated using a Stratagene Mx300P system (Stratagene, La Jolla, CA, USA).

## Examination of persister cells formation

In order to check likely formation of persister cells, first, MRSA or MSSA was grown in LB broth medium overnight. Then they were diluted by supernatants of *E. coli* 25922 to a density of $3*10^4$ cells/ml for 20 minutes and another 20 minutes with added 2 μg/ml rifampin at 37°C without shaking (to assess *E. coli* EDF plus rifampin efficacy). Finally, rifampin MIC for the remaining cells were evaluated by macro broth dilution method.

## Investigating the probable EDF production by MRSA and MSSA and its likely effect on themselves

For this aim, MRSA and MSSA were cultured in LB broth until their optical density at 600 nm [OD600] reached more than two. Then, the cultures were incubated with sub-MIC of rifampin at 37°C without shaking for 40 minutes. Finally, the loss of viability was determined by counting CFUs.

## Data analysis

In different parts of this study, the data was confirmed by replicating the tests. For example, we determined MIC by a duplicating method. Also, determination of EDF efficacy in different conditions was done while different steps were repeated several times. The evaluation of *mazEF* gens expression in MRSA and MSSA in different conditions was confirmed by duplicated method of Real-time PCR, as well (fold change was calculated for the genes as 2^(-ΔΔCT)).

The statistical analysis was performed using IBM SPSS statistical software, version 22.0 (SPSS Inc., Chicago, IL). To assess normality of the data, Kolmogorov–Smirnov test was performed. The data between the studied groups were compared using one-way ANOVA test. In all the analyses, $p < 0.05$ was considered statistically significant. The data were expressed as the mean value plus–minus the standard error of the mean (mean ± SEM).

## Results

### Identification and verification of bacterial strains

All of the bacterial strains (ATCC43300 and ATCC25923 as MRSA and MSSA standard strains, respectively, and ATCC25922 as the standard strain of *E. coli* and the clinical isolate of this bacterium) were confirmed using the CLSI 2022 guideline standard identification methods. The MRSA strain was resistant to cefoxitin (30 μg disk) while the MSSA was susceptible.

### Determination of the EDF efficacy

As described in the Methods, the *E. coli* strains were cultured in different conditions and their supernatants were collected and then the MRSA and MSSA strains were influenced by these supernatants and finally their CFUs were measured. When the MRSA and MSSA were affected by the standard EDF of *E. coli* 25922 (EDF of mid-logarithmic growth phase, OD = 0.6) plus sub-MIC rifampin (2 μg/ml), the CFUs were decreased over than 3 times (p<0.001). But the standard EDF of *E. coli* 25922 alone as well as sub-MIC rifampin alone, didn't have significant effect on the CFUs of the MRSA and MSSA strains (Fig 1A).

Similarly, the EDF derived from the clinical isolate of *E. coli* (from mid-logarithmic growth phase, OD = 0.6) plus sub-MIC rifampin decreased the CFUs of MRSA and MSSA almost 2 times (p<0.001). But the only EDF or sub-MIC rifampin alone didn't have significant effect on the CFUs of MRSA or MSSA (Fig 1B).

The supernatants of *E. coli* 25922 in different conditions (from different ODs) were evaluated. none of them could considerably decrease the CFUs of MRSA or MSSA except in case of OD = 0.78 which had a little effect in decreasing the CFUs of MRSA (p = 0.03) (Fig 2A).

As well, the supernatant of *E. coli* 25922 was extracted in LB broth (24h and 72h), M9 without glucose, and M9 without amino acids and evaluated. These supernatants also didn't decrease the CFUs of MRSA or MSSA in comparison to the supernatant of the mid-logarithmic growth phase (OD = 0.6) (Fig 2B).

### Real-Time PCR

The Real-Time PCR was applied to evaluate the expression of *mazE* and *mazF* genes. As a result, the expression of *mazF* gene in MRSA and MSSA when influenced by EDF (from either *E. coli* 25922 or the clinical strain) plus rifampin was highly more than the expression of *mazE* gene when compared to their normal growth conditions (without any treatment). Meanwhile, the expression of *mazE* and *mazF* genes in MRSA and MSSA influenced by only the EDF (from either *E. coli* 25922 or the clinical strain), didn't show significant difference with the expression of these genes in their normal growth conditions. Moreover, the expression of *mazF* gene in MRSA increased when affected by only sub-MIC rifampin (p<0.001); however, this effect was not detected in MSSA (Fig 3A and 3B).

### MIC determination

The MIC of rifampin for MRSA was 8 μg/ml and for MSSA was 16 μg/ml. Furthermore, MRSA and MSSA had no growth in 16 μg/ml and 32 μg/ml of rifampin, respectively.

### Formation of persister cells

The possible formation of persister cells from MRSA and MSSA that remained after they challenged with rifampin plus supernatant of *E. coli* 25922 was investigated. Surprisingly, the MIC of rifampin for MRSA and MSSA raised by 2 and 3 times, respectively when compared with their MIC in their normal growth conditions (without any challenging).

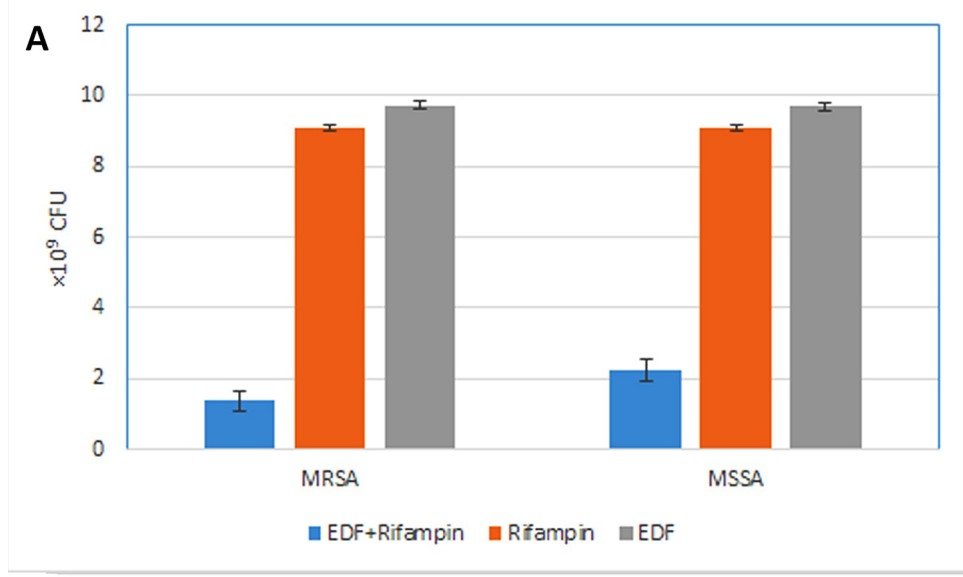

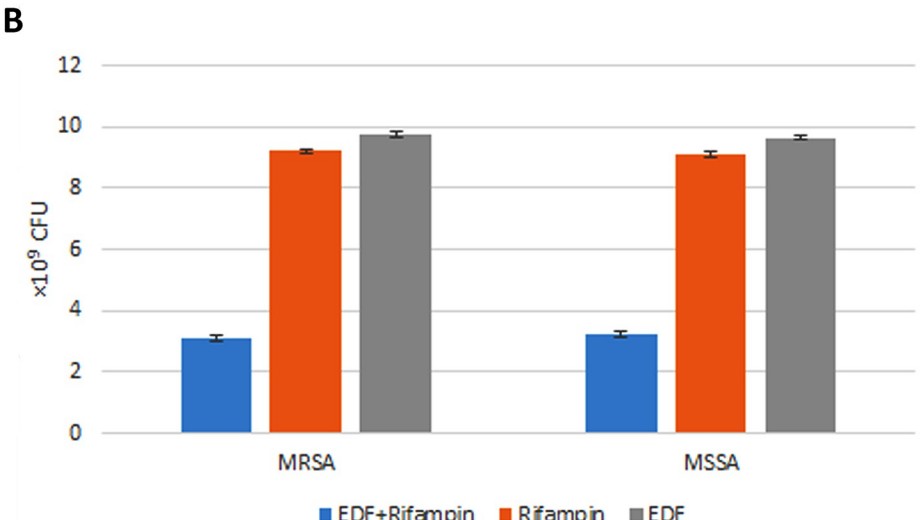

**Fig 1. The effect of EDF derived from *E. coli* (in mid-logarithmic growth phase, OD = 0.6) on the MRSA and MSSA. (A)** When the MRSA and MSSA were affected by the standard EDF of *E. coli* 25922 plus sub-MIC rifampin (2 μg/ml), the CFUs were decreased over than 3 times ($p<0.001$). But the standard EDF alone as well as sub-MIC rifampin alone, didn't have significant effect on the CFUs of the MRSA and MSSA strains; **(B)** The EDF derived from the clinical isolate of *E. coli* plus sub-MIC rifampin decreased the CFUs of MRSA and MSSA almost 2 times ($p<0.001$). But the only EDF or sub-MIC rifampin alone didn't have significant effect on the CFUs of MRSA or MSSA. **Note:** $10*10^9$ CFU considered as the complete growth. In all the analyses, $p<0.05$ was considered statistically significant. The data were expressed as the mean value plus–minus the standard error of the mean (mean ± SEM).

### EDF production by MRSA and MSSA and its effect on themselves

The result of the experiment showed more than two times decreasing in the CFUs of the MRSA and MSSA when the cultures were challenged with sub-MIC rifampin ($p<0.001$) (Fig 2C).

### Discussion

Today, bacterial infections are the great part of healthcare problems in the world and annually they cause many dangerous diseases and even death. One of the important bacteria capable to

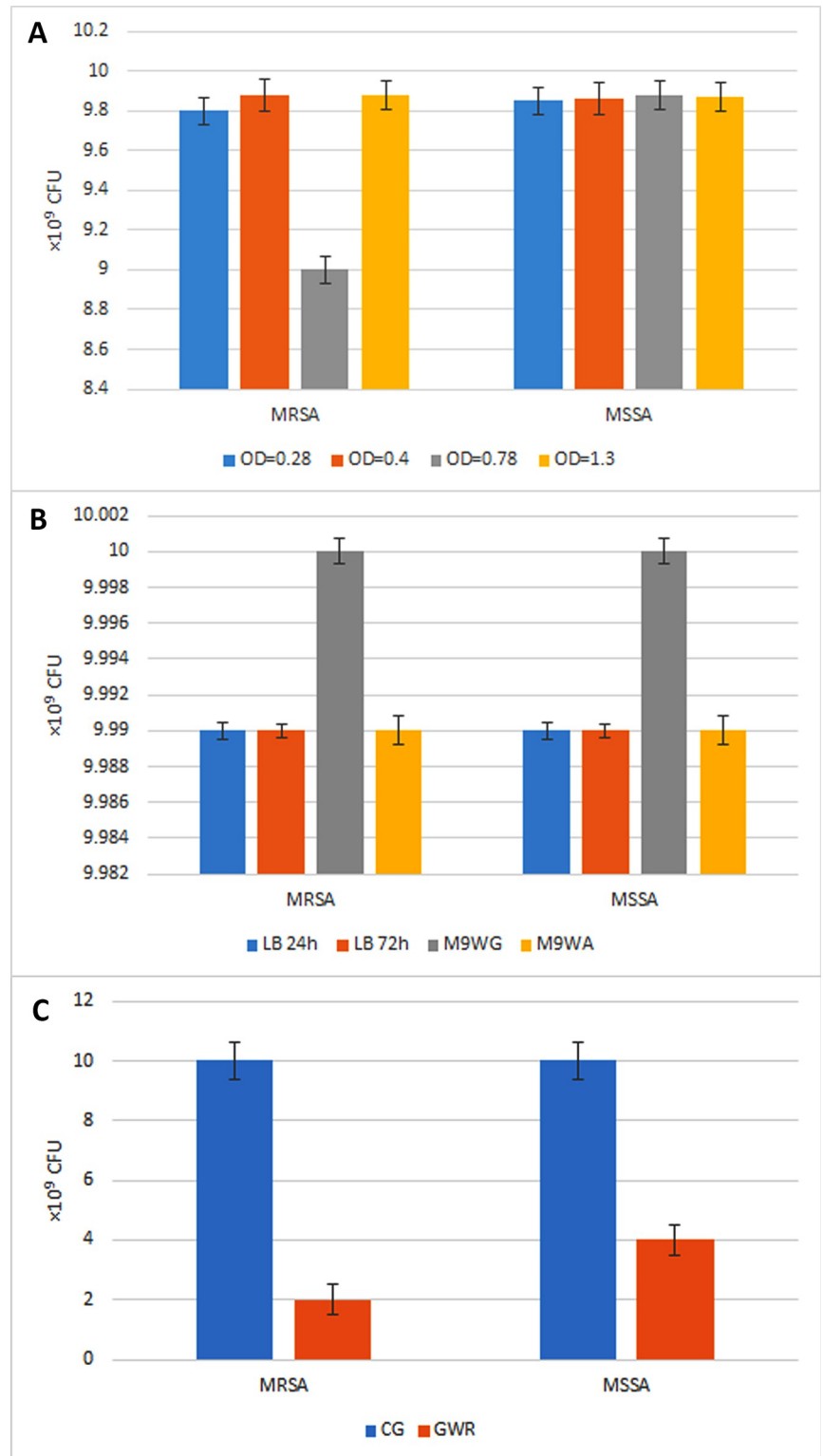

**Fig 2. The effect of the supernatants of *E. coli* 25922 in different conditions on MRSA and MSSA and their self-production of EDF.** **(A)** The supernatant from different ODs (OD = 0.28, OD = 0.4, OD = 0.78 and OD = 1.3): none of them could considerably decrease the CFUs of MRSA or MSSA except in case of OD = 0.78 which had a little effect in decreasing the CFUs of MRSA (p = 0.03). **(B)** The supernatant extracted when the bacterium grown in LB broth (24h and 72h), M9 without glucose, and M9 without amino acids. These supernatants also didn't decrease the CFUs of

MRSA or MSSA in comparison to the supernatant of the mid-logarithmic growth phase (OD = 0.6). **(C)** Investigating the probable production of EDF by MRSA and MSSA and its likely effect on themselves: The result showed more than two times decreasing in the CFUs of the MRSA and MSSA when the cultures were challenged with sub-MIC rifampin (p<0.001). **Note:** $10*10^9$ CFU considered as the complete growth. CG: complete growth; GWR: growth with rifampin (sub-MIC); M9WG: M9 without glucose; M9WA: M9 without amino acids. In all the analyses, p<0.05 was considered statistically significant. The data were expressed as the mean value plus–minus the standard error of the mean (mean ± SEM).

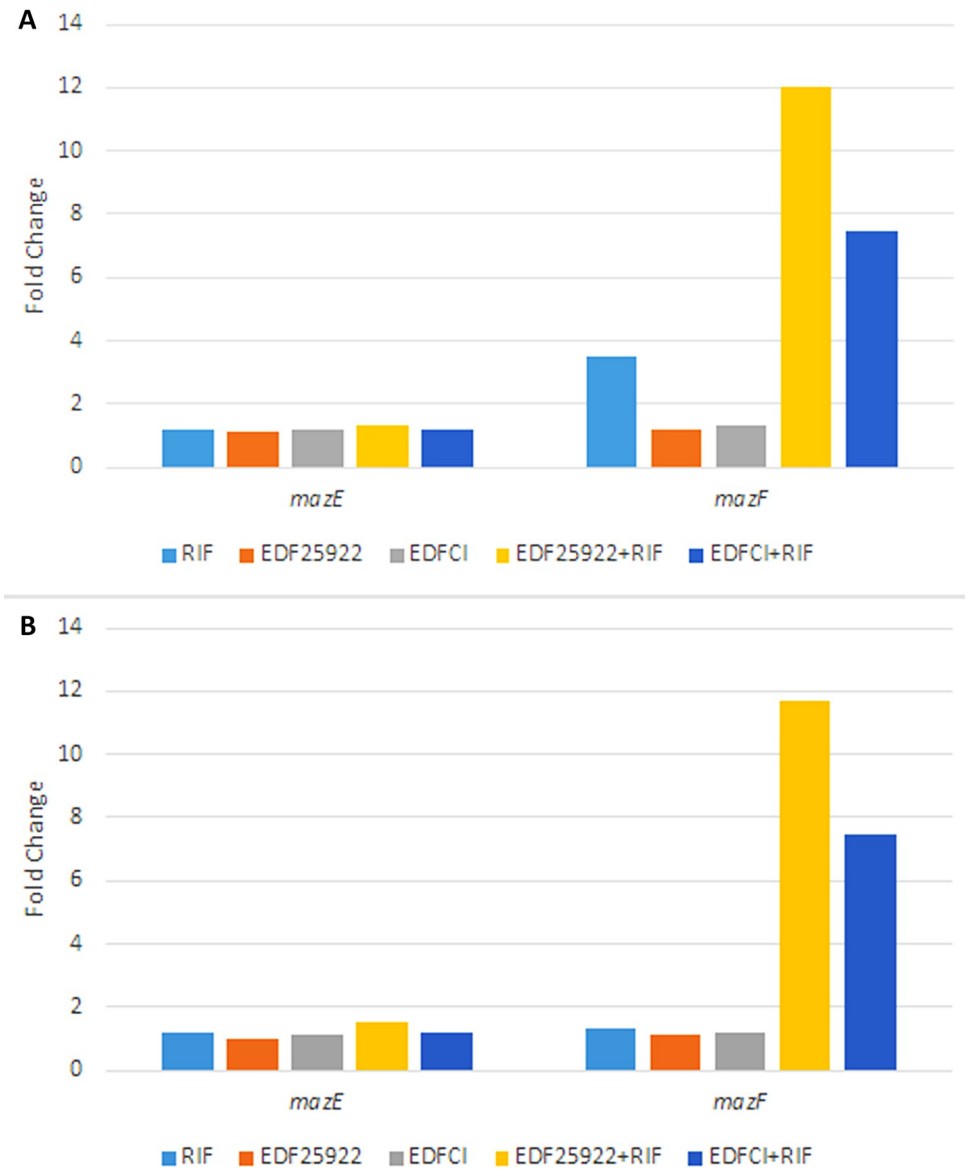

**Fig 3. Evaluating the expression of *mazE* and *mazF* genes by Real-time PCR.** The expression of *mazF* gene in MRSA **(A)** and MSSA **(B)** when influenced by EDF (from either *E. coli* 25922 or the clinical strain) plus rifampin was highly more than the expression of *mazE* gene when compared to their normal growth conditions (without any treatment). Moreover, the expression of *mazF* gene in MRSA (but not in MSSA) increased when affected by only sub-MIC rifampin (p<0.001). RIF: rifampin; EDF25922: EDF derived from *E. coli* 25922 standard strain; EDFCI: EDF derived from clinical isolate of *E. coli*. Fold change was calculated for the genes as 2^(-ΔΔCT). In all the analyses, p<0.05 was considered statistically significant.

cause various infections is *S. aureus*, although it may exist as one of the normal microbiota in humans [11].

Nowadays, methicillin-resistant and methicillin-sensitive *S. aureus* (MRSA and MSSA) are medically important bacteria capable to create significant infections. Since MRSA was described in 1961, it has been one of the key bacteria which has caused healthcare- as well as community-associated infections [12]. The increasing rate of healthcare-associated infections caused by MRSA particularly in developing countries has led to many public health problems. MRSA isolates are usually resistant to the majority of the antibiotics commonly used to treat the human infections caused by *S. aureus*. It is an important cause of severe nosocomial infections and a major public health concern worldwide [13].

As a fact, the resistant bacteria are more dangerous but the persistent ones could be much more dangerous. The persister bacteria are remarkably resistant to conventional doses of antibiotics and they have ability to cause very important problems such as recurrent infections and increasing of treatment duration. Although these bacteria are extremely resistant to antibiotics, but genetically they are identical to the other cells. Thus, they are phenotypic variants and are completely different from resistant bacteria. The first detection of bacterial persister cell formation was in the late 1944 in *Staphylococcus aureus*. The formation of persister cells has been detected in many pathogenic bacteria and among them, the persister cell formation in *S. aureus* could be one of the important global threatening problems [14–16]. Unfortunately, because of their high resistances to antibiotics, persister cells are hardly eradicated [17, 18]. Therefore, it seems that the novel practical methods are needed to combat the persister cells.

Adaptability and survival are two important issues for all living organisms, including bacteria. These microorganisms have the ability to control cell death under several stressful conditions, such as nutrient deprivation, radiation exposure, high temperature, phage infections, oxidative stress, antibacterial pressure, and DNA damage [19]. When the bacteria undergo cellular damage or some stressing conditions, the TA system may help the bacteria to survive and pass the crisis resulting in either cell growth arrest or a form of cell death similar to apoptosis [20]. In most of these cases, the TA system may induce PCD in a part of the bacterial population to survive the colony as a result of "population based behaviors" occurring in the bacterial communities [19].

EDF, a linear pentapeptide, is a quorum sensing factor capable to induce MazEF-mediated cell death in *E. coli* trough amplifying the enzymatic activity of the toxin MazF [7].

It has been recently evidenced that the EDFs of *B. subtilis* and *P. aeruginosa* are able to trigger the *E. coli* MazEF system. This may indicate an altruistic suicide (self-sacrifice) mechanism capable to be used in elimination of the competitor species in a mixed bacterial population [8]. Therefore, EDFs may have the potential to be exploited as a new class of antimicrobial agents in order to trigger cell death from outside the bacterial cells.

Consistently, the results of the current study showed that the *E. coli* EDF could significantly reduce the CFUs of MRSA and MSSA. This may be a promising tool to combat the infections caused by the antibiotic-resistant pathogens, as a non-antibiotic treatment candidate.

In a study, Kolodkin-Gal and Engelberg-Kulka showed that EDF production of various *E. coli* strains is different [6]. In the present study, we used two different *E. coli* strains, (*E. coli* 25922 and the clinical strain), to produce EDF available from their supernatants. Our results indicate that the supernatant of each of these two *E. coli* strains had different effect on MRSA and MSSA CFUs in which the supernatant of *E. coli* 25922 was more powerful than the clinical strain ones to decrease the CFUs of the MRSA and MSSA.

As mentioned in Methods, we followed the standard mentioned in some studies to produce EDF from the *E. coli* strains [6, 8]. However, in our study we tried to produce the supernatants (as EDFs) in different conditions. Therefore, we extracted the supernatant of *E. coli* 25922 not

only in mid-logarithmic growth phase (OD = 0.6) but also in OD = 0.28, OD = 0.4, OD = 0.78, OD = 1.3, LB broth (24h and 72h), M9 without glucose, as well as in M9 without amino acids. As we expected, these different supernatants did not decrease the CFUs of MRSA or MSSA in comparison to the supernatant of the mid-logarithmic growth phase (OD = 0.6). This finding indicates that the EDF is a QS peptide that is produced in a specific condition.

Researchers have revealed that the EDF is necessary for initiation of PCD and moreover, the EDF itself needs a trigger such as sub-MIC rifampin to function [21, 22]. Accordingly, we used sub-MIC rifampin to stimulate EDF to initiate the *mazEF* mediated cell death and we showed that the EDF alone was not able to induce PCD by the mediation of *mazEF* system.

The previous studies have shown that EDF could control *mazEF* genes expression and consequently, PCD is occurred by the involvement of *mazEF* system [21, 22]. In this process, the EDF increases the *mazF* gene expression. As we showed here, when MRSA or MSSA was challenged with both the EDF (from either *E. coli* 25922 or the clinical strain) and rifampin, the expression of *mazF* was highly increased. Furthermore, the EDF alone (from either *E. coli* 25922 or the clinical strain) could not increase the expression of *mazF* in EDF-challenged MRSA or MSSA strains. These findings definitely indicate that the EDF in combination with its trigger (rifampin) could cause PCD through activation of the MazEF system.

Kumar *et al.* discovered that mazEF-mediated cell death in *E. coli* could be triggered by QS peptides from the supernatants of the Gram-positive bacterium *Bacillus subtilis* as well as the Gram-negative bacterium *Pseudomonas aeruginosa* [8]. Here, for the first time, we extracted the supernatants of two aforementioned *E. coli* strains as EDF and evaluated their effects in elimination of MRSA and MSSA standard strains. Therefore, we consistently and affirmatively might conclude that EDF yielded from the different bacterial species could cause PCD in the other bacteria. The increasing rate of antibiotic resistance in many bacterial pathogens highlights the use of non-antibiotic therapeutic strategies to overcome the pathogens as well as to reduce or avoid the risk of side effects caused by different antibiotics. In this regard, the exploiting of EDF for PCD induction, may be a useful as a non-antibiotic therapeutic approach, to fight against the antibiotic-resistant bacteria capable to cause the chronic and persistent infections that are the global crisis. However, this in vitro technique should be more evaluated using animal model studies as well as the clinical trials in the future.

Nevertheless, a small sub-population of bacteria (perhaps the persister cells) may survive and likely become the nucleus of a new population by removal of EDF effect or when the growth conditions become less stressful [23]. The survived sub-population of pathogens could develop and renew the colony which may cause the recurrent and persistent infection that is of high importance. Therefore, these fraction of bacteria should be considered when we try to eliminate and eradicate the infection by means of this phenomenon.

In the present study, we also investigated two theories: 1) the likely formation of persister cells from MRSA and MSSA, and 2) the possible synthesis of EDF in old cultures of these pathogens. The recent studies revealed that there is a communication between TA system and persister cells [24]. Moreover, the combination of rifampin and EDF could activate MazEF TA system [6, 8]. Subsequently, one may conclude that when TA system is active, the persister cells could be formed. About this theory, as mentioned in the Methods, we saw an unusual increasing of rifampin MIC in MRSA and MSSA strains that remained after they challenged with sub-MIC rifampin plus the EDF of *E. coli* 25922. Therefore, maybe these resistant remaining bacteria are persister cells although the more researches are needed to confirm this finding.

About the second theory, many studies have shown that EDF is a QS molecule that depends on bacterial population density [6, 25]. Thus, the EDF is produced in overpopulation. When EDF is produced in the bacterial population and rifampin (as a stress stimulating factor) is added, the PCD (*mazEF* mediated cell death) is occurred and most of the bacterial population

is destroyed [8]. In the present study, sub-MIC rifampin was added to the old broth cultures of MRSA and MSSA. Surprisingly, the CFUs of these pathogens significantly decreased indicating that EDF has been produced in the old cultures of MRSA and MSSA.

It should be mentioned that this research, however, is subject to several limitations. For example, we would like to characterize and purify EDF of each *E. coli* strain from the related supernatants but unfortunately we did not have access to sufficient funds or the necessary tools to perform. Moreover, we wanted to provide a synthetic EDF of *E. coli* and evaluate its effect on the CFUs of MRSA and MSSA, but we could not do that because of some limitations including the financial inadequacy.

Finally, although the exploitation of the communication between EDF and PCD to fight against the pathogens is encouraging, more related studies are required to develop and apply this phenomenon as a potential antibacterial tool against bacterial pathogens. Furthermore, the possible survival of small sub-population of the bacteria and the safety and ethical implications of this therapeutic strategy should be considered and precisely investigated.

## Conclusion

Our findings indicate that the EDF derived from *E. coli* in combination with sub-MIC rifampin could induce PCD in MRSA and MSSA through activation of the MazEF system. This may be exploited as a non-antibiotic treatment candidate to combat the infections caused by the antibiotic-resistant pathogens capable to cause chronic and persistent infections that are the global crisis. However, this *in vitro* technique should be more evaluated using animal model studies as well as the clinical trials in the future.

## Supporting information

**S1 File. Supporting graphs for the role of the EDF derived from *E. coli* in the induction of programmed cell death (PCD) in MRSA and MSSA.**
(RAR)

## Author Contributions

**Conceptualization:** Mohammad Hossein Ahmadi.

**Data curation:** Shahriar Bakhti.

**Formal analysis:** Shahriar Bakhti, Parviz Owlia.

**Funding acquisition:** Mohammad Hossein Ahmadi.

**Investigation:** Shahriar Bakhti, Mohammad Hossein Ahmadi, Parviz Owlia.

**Methodology:** Shahriar Bakhti, Mohammad Hossein Ahmadi, Parviz Owlia.

**Project administration:** Mohammad Hossein Ahmadi.

**Software:** Shahriar Bakhti.

**Supervision:** Mohammad Hossein Ahmadi.

**Validation:** Mohammad Hossein Ahmadi, Parviz Owlia.

**Writing – original draft:** Shahriar Bakhti.

**Writing – review & editing:** Mohammad Hossein Ahmadi, Parviz Owlia.

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
