## [Decision Letter · Decision Letter 0]

17 Oct 2024

PONE-D-24-30735Inducing programmed cell death trough MazEF system to combat Staphylococcus aureus: a non-antibiotic treatment optionPLOS ONE

Dear Dr. Ahmadi,

Thank you for submitting your manuscript to PLOS ONE. After careful consideration, we feel that it has merit but does not fully meet PLOS ONE’s publication criteria as it currently stands. Therefore, we invite you to submit a revised version of the manuscript that addresses the points raised during the review process.

We look forward to receiving your revised manuscript.

Kind regards,

Hideo Kato

Academic Editor

PLOS ONE

**Journal Requirements:**

This work is based upon research funded by Iran National Science Foundation (INSF) under project No. 4002145.

Reviewers' comments:

Reviewer's Responses to Questions

**Comments to the Author**

1. Is the manuscript technically sound, and do the data support the conclusions?

Reviewer #1: Yes

Reviewer #2: Yes

Reviewer #3: Yes

Reviewer #4: Yes

2. Has the statistical analysis been performed appropriately and rigorously? 

Reviewer #1: Yes

Reviewer #2: Yes

Reviewer #3: Yes

Reviewer #4: Yes

3. Have the authors made all data underlying the findings in their manuscript fully available?

Reviewer #1: Yes

Reviewer #2: Yes

Reviewer #3: Yes

Reviewer #4: Yes

4. Is the manuscript presented in an intelligible fashion and written in standard English?

Reviewer #1: Yes

Reviewer #2: Yes

Reviewer #3: Yes

Reviewer #4: Yes

5. Review Comments to the Author

**Reviewer #1:** The manuscript is interesting and scientifically sounding. The introduction is clear and comprehensive, the methods are extremely detailed, allowing the replication of the performed experiments, and report a good statistical analysis. The results are well commented in the discussion and the main objective of the study. Still, there are some concerns that need to be addressed:

1. The induction of programmed cell death trough MazEF system to combat Staphylococcus aureus is not currently an option but it may be a candidate. Thus, I suggest the authors to replace "option" with "candidate" in the title of the manuscript and correct the title as: "Inducing programmed cell death trough MazEF system to combat Staphylococcus aureus: a non-antibiotic treatment candidate".

2. Please also replace "option" with "candidate" throughout the manuscript text, where applicable.

3. Why the authors used M9 minimal medium with 1% glucose to extract the EDF from E. coli?

4. Why the only EDF or sub-MIC rifampin alone didn’t have significant effect on the CFUs of MRSA or MSSA in your study?

5. In this study., the supernatant of E. coli 25922 was extracted in LB broth (24h and 72h), M9 without

glucose, and M9 without amino acids. The authors declare that these supernatants didn’t decrease

the CFUs of MRSA or MSSA in comparison to the supernatant of the mid-logarithmic growth

phase. What could be the reason?

**Reviewer #2: **The interesting topic, The study has well disigned, well performed and the manuscript well written.

Due to the increasing resistance rate to conventional thraprutic method and antibiotics, this is very important to seek for novel treatments and new alternatives for the antibiotics.

**Reviewer #3:** The utility of conventional antibiotics for the treatment of bacterial infections has become increasingly strained due to increased rates of resistance coupled with reduced rates of development of new agents. As a result, multidrug-resistant, extensively drug-resistant, and pan-drug-resistant bacterial strains are now frequently encountered.

Alternatives therapeutic approaches to conventional antibiotics need to be explored to ensure that a robust pipeline of effective therapies is available to clinicians. This study introduces a novel non-antibiotic solution to manage invasive bacterial infections.

**Reviewer #4: **Thank you for giving me the opportunity to review this article. In my opinion, the article is scientifically strong, it studies an important topic, it has a good design, and it is well executed. I think this article is suitable for the journal.

6. PLOS authors have the option to publish the peer review history of their article (what does this mean?). If published, this will include your full peer review and any attached files.

Reviewer #1: No

Reviewer #2: No

Reviewer #3: No

Reviewer #4: No

---

## [Author Response · Author response to Decision Letter 0]

3 Nov 2024

Dear Dr. Hideo Kato

Academic Editor

PLOS ONE

We highly appreciate the editor and anonymous reviewers for their constructive comments which helped us to improve the manuscript. We do appreciate the opportunity to respond to the comments by the journal honorable reviewers. 

As suggested, we have revised the manuscript and responded to the reviewers’ comments. Moreover, we have addressed all issues raised by the dear Academic Editor. In this letter, I will briefly summarize the important changes we have made in the manuscript and also respond to the reviewer’s comments.

Respectfully,

Mohammad Hossein Ahmadi

Corresponding Author

Reviewer #1: The manuscript is interesting and scientifically sounding. The introduction is clear and comprehensive, the methods are extremely detailed, allowing the replication of the performed experiments, and report a good statistical analysis. The results are well commented in the discussion and the main objective of the study. Still, there are some concerns that need to be addressed:

1. The induction of programmed cell death trough MazEF system to combat Staphylococcus aureus is not currently an option but it may be a candidate. Thus, I suggest the authors to replace "option" with "candidate" in the title of the manuscript and correct the title as: "Inducing programmed cell death trough MazEF system to combat Staphylococcus aureus: a non-antibiotic treatment candidate".

Our comments: Very thanks for your good comments. As suggested, we have replaced "option" with "candidate" in the title of the manuscript and corrected the title as: "Inducing programmed cell death trough MazEF system to combat Staphylococcus aureus: a non-antibiotic treatment candidate".

2. Please also replace "option" with "candidate" throughout the manuscript text, where applicable.

Our comments: As suggested, we have replaced "option" with "candidate" throughout the manuscript text, where applicable.

3. Why the authors used M9 minimal medium with 1% glucose to extract the EDF from E. coli?

Our comments: As mentioned in the referenced articles, under the “minimal growth condition”, the production of EDF increases and PCD begins. Therefore, the M9 medium has been used to create such conditions in various articles, and we also used this environment in this study.

4. Why the only EDF or sub-MIC rifampin alone didn’t have significant effect on the CFUs of MRSA or MSSA in your study?

Our comments: The MazEF-mediated cell death could be induced by stressful conditions such as the exposure to the rifampin, moreover, the mazEF-mediated cell death is a population phenomenon requiring a quorum-sensing factor called extracellular death factor (EDF). So, it seems that both rifampin and EDF have synergistic effect together.

5. In this study., the supernatant of E. coli 25922 was extracted in LB broth (24h and 72h), M9 without glucose, and M9 without amino acids. The authors declare that these supernatants didn’t decrease the CFUs of MRSA or MSSA in comparison to the supernatant of the mid-logarithmic growth phase. What could be the reason?

Our comments: As mentioned, the EDF as a quorum-sensing factor is produced in stressful conditions. One of the important stressful conditions could be the starvation. It seems that the most production of EDF is only in M9 medium with amino acids as a minimal medium.

Reviewer #2: The interesting topic, the study has well disigned, well performed and the manuscript well written. Due to the increasing resistance rate to conventional thraprutic method and antibiotics, this is very important to seek for novel treatments and new alternatives for the antibiotics.

Our comments: Very thanks for your comments.

Reviewer #3: The utility of conventional antibiotics for the treatment of bacterial infections has become increasingly strained due to increased rates of resistance coupled with reduced rates of development of new agents. As a result, multidrug-resistant, extensively drug-resistant, and pan-drug-resistant bacterial strains are now frequently encountered.

Alternatives therapeutic approaches to conventional antibiotics need to be explored to ensure that a robust pipeline of effective therapies is available to clinicians. This study introduces a novel non-antibiotic solution to manage invasive bacterial infections.

Our comments: Very thanks for your comments.

Reviewer #4: Thank you for giving me the opportunity to review this article. In my opinion, the article is scientifically strong, it studies an important topic, it has a good design, and it is well executed. I think this article is suitable for the journal.

Our comments: Very thanks for your comments.

---

## [Decision Letter · Decision Letter 1]

19 Nov 2024

Inducing programmed cell death trough MazEF system to combat Staphylococcus aureus: a non-antibiotic treatment candidate

PONE-D-24-30735R1

Dear Dr. Ahmadi,

We’re pleased to inform you that your manuscript has been judged scientifically suitable for publication and will be formally accepted for publication once it meets all outstanding technical requirements.

Kind regards,

Hideo Kato

Academic Editor

PLOS ONE

Additional Editor Comments (optional):

Reviewers' comments:

Reviewer's Responses to Questions

**Comments to the Author**

1. If the authors have adequately addressed your comments raised in a previous round of review and you feel that this manuscript is now acceptable for publication, you may indicate that here to bypass the “Comments to the Author” section, enter your conflict of interest statement in the “Confidential to Editor” section, and submit your "Accept" recommendation.

Reviewer #1: All comments have been addressed

Reviewer #2: All comments have been addressed

Reviewer #3: All comments have been addressed

Reviewer #4: All comments have been addressed

2. Is the manuscript technically sound, and do the data support the conclusions?

Reviewer #1: Yes

Reviewer #2: Yes

Reviewer #3: Yes

Reviewer #4: Yes

3. Has the statistical analysis been performed appropriately and rigorously? 

Reviewer #1: Yes

Reviewer #2: Yes

Reviewer #3: Yes

Reviewer #4: Yes

4. Have the authors made all data underlying the findings in their manuscript fully available?

Reviewer #1: Yes

Reviewer #2: Yes

Reviewer #3: Yes

Reviewer #4: Yes

5. Is the manuscript presented in an intelligible fashion and written in standard English?

Reviewer #1: Yes

Reviewer #2: Yes

Reviewer #3: Yes

Reviewer #4: Yes

6. Review Comments to the Author

Reviewer #1: The authors have adequately addressed all of my comments and I think that the manuscript is now acceptable.

Reviewer #2: (No Response)

Reviewer #3: I think all comments have been addressed by the authors. I have already recommended accepting this article.

Reviewer #4: (No Response)

7. PLOS authors have the option to publish the peer review history of their article (what does this mean?). If published, this will include your full peer review and any attached files.

Reviewer #1: No

Reviewer #2: No

Reviewer #3: No

Reviewer #4: No

---

## [Editor Report · Acceptance letter]

26 Nov 2024

PONE-D-24-30735R1 

PLOS ONE

Dear Dr. Ahmadi, 

I'm pleased to inform you that your manuscript has been deemed suitable for publication in PLOS ONE. Congratulations! Your manuscript is now being handed over to our production team.

Kind regards, 

on behalf of

Dr. Hideo Kato 

Academic Editor

PLOS ONE